# Hybrid CtrlFormer: Learning Adaptive Search Space Partition for Hybrid Action Control via Transformer-based Monte Carlo Tree Search

**Jiashun Liu**[*1]     **Xiaotian Hao**[*1]     **Jianye Hao**[†1]     **Yan Zheng**[1]     **Yujing Hu**[2]     **Changjie Fan**[2]     **Tangjie Lv**[2]

**Zhipeng Hu**[2]

[1]Tianjin University, Tianjin, China
[2]FUXI AI Laboratory, NetEase, Hangzhou, China

## Abstract

Hybrid action control tasks are common in the real world, which require controlling some discrete and continuous actions simultaneously. To solve these tasks, existing Reinforcement Learning (RL) methods either directly build a separate policy for each type of action or simplify the hybrid action space into a discrete or a continuous action control problem. However, these methods neglect the challenge of exploration resulting from the complexity of the hybrid action space. Thus, it is necessary to design more sample efficient algorithms. To this end, we propose a novel Hybrid Control Transformer (Hybrid CtrlFormer), to achieve better exploration and exploitation for the hybrid action control problems. The core idea is: ① we construct a hybrid action space tree with the discrete actions at the higher level and the continuous parameter space at the lower level. Each parameter space is split into multiple subregions. ② To simplify the exploration space, a Transformer-based Monte-Carlo tree search method is designed to efficiently evaluate and partition the hybrid action space into good and bad subregions along the tree. Our method achieves state-of-the-art performance and sample efficiency in a variety of environments with discrete-continuous action space.

## 1 INTRODUCTION

The field of deep reinforcement learning (DRL) has witnessed striking empirical achievements in a variety of Markov Decision Process (MDP) problems, involving controls with either discrete actions, such as Go [Silver et al., 2016], or continuous actions, such as robot control [Schulman et al., 2015, Lillicrap et al., 2016]. However, many real-world scenarios require more complex controls with discrete-continuous hybrid action spaces, which are usually modeled as Parameterized Action Markov Decision Processes (PAMDP) [Hausknecht and Stone, 2016, Massaroli et al., 2020]. Typical discrete-continuous hybrid control tasks include Platform [Masson et al., 2016], Move [Li et al., 2021] and Robot Soccer [Xiong et al., 2018]. Take Pac-man as an example (Figure 1 A), the agent has to first choose a discrete action (i.e., *move*, *catch*), and then set the corresponding continuous parameters for the selected discrete action (i.e., *radian*). Compared with the discrete or continuous action space, the exploration space of hybrid action control tasks becomes a complex hierarchy. However, previous DRL methods designed either for discrete or continuous action spaces cannot be directly applied to such heterogeneous (discrete-continuous hybrid) control problems.

To solve the hybrid control problems, three types of methods have been proposed. The first type of methods simply convert the heterogeneous action space into a homogeneous one by either discretizing all continuous actions ([Massaroli et al., 2020]) or relaxing all discrete actions into continuous ones [Hausknecht and Stone, 2016, Masson et al., 2016]. However, discretizing all dimensions of the continuous action suffers from the loss of control accuracy and the scalability issue; while casting all discrete actions into continuous ones enlarges the original action space, resulting in additional difficulties in approximation and generalization[Li et al., 2021]. The second type of methods directly build separate policies for the discrete actions and the continuous ones. For example, Hybrid PPO (HPPO) [Fan et al., 2019] builds multiple network heads to learn the hybrid policy. However, building a separate continuous action policy network for each discrete action will enlarge the model size, resulting in additional difficulties in optimization.

To improve the learning efficiency, the third type of methods explicitly incorporate the dependencies between the discrete actions and the continuous parameters into the model design. Parameterized DQN (PDQN) [Xiong et al., 2018]

---

\* equal contribution, † corresponding author

*Accepted for the 40th Conference on Uncertainty in Artificial Intelligence* (UAI 2024).

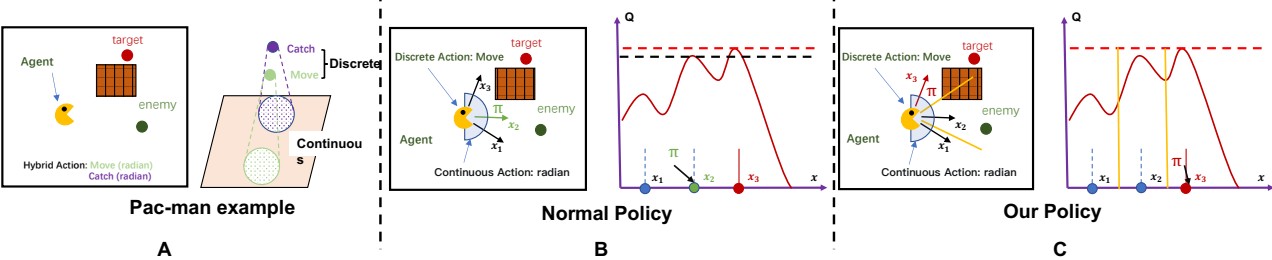

Figure 1: Illustration of the Pac-man game and the landscape of the Q-values. The target of the agent (in yellow) is to eat the red point without touching the wall or enemy.

proposes a hybrid structure by combining DQN [Silver et al., 2016] with DDPG [Lillicrap et al., 2016]. Thus, the continuous action policy is directly affected by the high-level discrete action policy. However, the dependence of PDQN's Q-values on all continuous actions causes a *false gradients* issue and can lead to suboptimal action selection. To address the *false gradients* issue, [Bester et al., 2019] further design a Multi-Pass Q-Network (MP-DQN), making each discrete action's Q-value only depends on its corresponding continuous parameters. Further, Fu et al. [2019] extends PDQN to multi-agent RL settings. Most recently, Li et al. [2021] propose Hybrid Action Representation (HyAR), which uses an embedding table and a conditional Variational Auto-Encoder (VAE) to convert both the discrete and the continuous actions into a more compact latent space. HyAR achieves state-of-the-art (SOTA) performance on typical hybrid control tasks with higher learning efficiency, especially for high-dimensional action spaces. However, the embedding table and the VAE have to be periodically retrained with the policy updating, which makes HyAR suffer from training instability. We present a detailed related work overview in **Appendix A**.

Apart from the action-dependency modeling, another challenge affecting learning efficiency is the hard exploration problem. As shown in Figure 1 B: At the current state, the agent is blocked by the wall and cannot catch the target directly. Even if the agent correctly selects the *move* action, it's still hard to acquire the *best-moving radian* due to the insufficiently explored continuous action space. How to efficiently explore and prune the search space has not been sufficiently studied, especially when the hybrid action space becomes high-dimensional.

In this paper, we propose Hybrid Control Transformer (Hybrid CtrlFormer), an efficient search space partition algorithm balancing the exploration and exploitation in the hybrid action space. The high-level idea is: (1) we construct a hybrid action space tree with the discrete actions at the higher-level and the continuous parameter space at the lower-level. (2) For each discrete action, we learn a Transformer-based Q-function to partition the corresponding continuous parameter space into good and bad subregions (with high

and low Q-values) along the tree, according to which we can quickly prune the high-dimensional search space and get a smaller one with better value. (3) Then, Monte Carlo tree search (MCTS) method is used to select discrete action and its corresponding continuous action subregion, which effectively balances exploration and exploitation. (4) A more fine-grained continuous control policy is trained only within the selected subregion, where the search space is significantly reduced.

Hybrid CtrlFormer can easily learn multi-modal policies which can potentially help escape from the sub-optimal policies. As shown in Figure 1 C, the continuous action space is divided into several subregions (the dividing line is visualized in yellow). Our policy can identify the appropriate subregion to simplify the action space and search for the optimal action (approaching the target without touching the wall or enemy). We evaluate Hybrid CtrlFormer in a variety of hybrid control tasks. The results demonstrate the superiority of Hybrid CtrlFormer in both the learning efficiency and the performance compared with existing baselines. Besides, Hybrid CtrlFormer can be easily extended to multi-agent settings by integrating it into existing value decomposition MARL methods [Rashid et al., 2018, Peng et al., 2021].

## 2 BACKGROUND

**Parameterized Action MDP (PAMDP).** In this paper, we focus on the PAMDP [Masson et al., 2016], which has a discrete-continuous hybrid action space and is an extension of MDP. PAMDP can be represented as: $(\mathcal{S}, \mathcal{H}, \mathcal{P}, \mathcal{R}, \gamma, T)$, where $\mathcal{H} = \{(k, x_k) | x_k \in \mathcal{X}_k, \forall k \in \mathcal{K}\}$, $\mathcal{K} = \{1, \cdots, K\}$ denotes the discrete action set, $\mathcal{X}_k$ is the corresponding continuous parameter set for each $k \in \mathcal{K}$. Each pair of $(k, x_k)$ constitutes a hybrid action. The state transition function and reward function are defined as $\mathcal{P} : \mathcal{S} \times \mathcal{H} \times \mathcal{S} \to [0, 1]$ and $\mathcal{R} : \mathcal{S} \times \mathcal{H} \to \mathbb{R}$ respectively. The agent's policy is defined as $\pi : \mathcal{S} \times \mathcal{H} \to [0, 1]$ and the state-action value function is denoted as $Q^\pi(s, k, x_k)$.

**Self-Attention in Transformer.** Self-attention module enables efficient modeling of data by capturing the interrela-

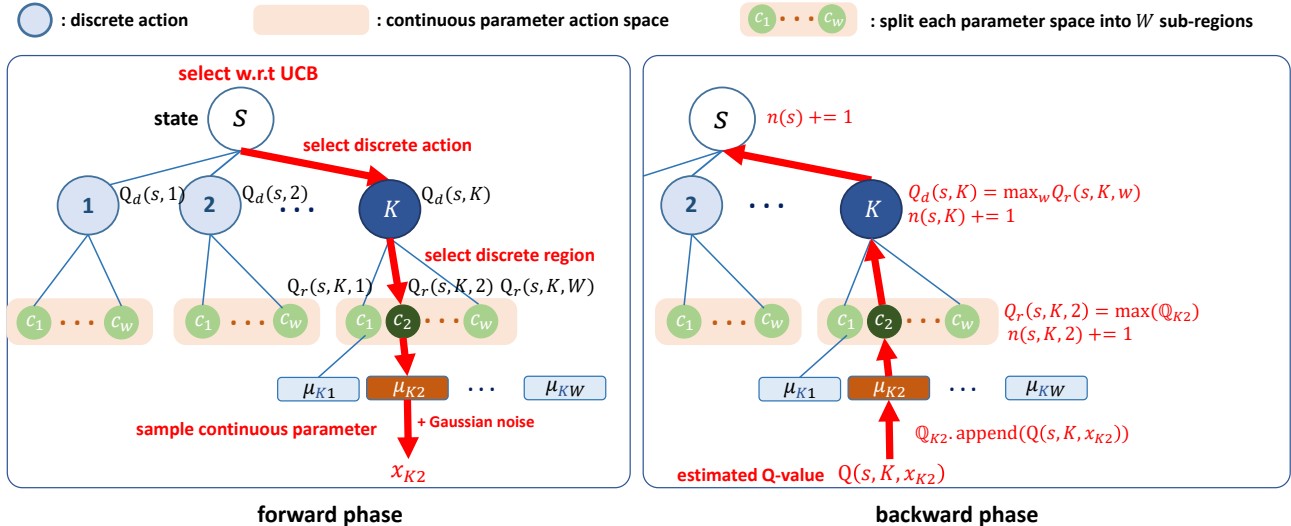

Figure 2: The overall architecture: learning to Split the Hybrid Action Space into Good and Bad subregions via MCTS.

tionship of input sequences. Assume that we have $n$ query vectors $\mathbb{Q} \in \mathbb{R}^{n \times d_q}$ and each with dimension $d_q$. The attention function maps queries $\mathbb{Q}$ to outputs using $n_v$ key-value pairs $\mathbb{K} \in \mathbb{R}^{n_v \times d_q}$ and $\mathbb{V} \in \mathbb{R}^{n_v \times d_q}$. The whole process can be written as $Attention(\mathbb{Q}, \mathbb{K}, \mathbb{V}) = Atten(\mathbb{Q}\mathbb{K}^T)\mathbb{V}$. The pairwise dot product $\mathbb{Q}\mathbb{K}^T$ reflects how similar each pair of query and key vectors is. $Atten(\mathbb{Q}\mathbb{K}^T)\mathbb{V}$ is a weighted sum of $\mathbb{V}$ where a value gains higher dynamic weight when its corresponding key has a larger dot product with the query. In this paper, we use a Transformer to measure the importance of each partitioned subregion.

**Monte-Carlo Tree Search.** Monte-Carlo Tree Search [Browne et al., 2012], or MCTS, is a heuristic search algorithm. In our setup, MCTS is used to find the optimal discrete action in the current state and its corresponding optimal continuous action subregion. In MCTS, each tree node X stores a value $v_X$ representing its goodness, and the number $n_X$ that it has been visited. They are used to calculate UCB [Rosin, 2011], i.e.,

$$v_X + c\sqrt{2(\ln n_p)/n_x} \quad (1)$$

where $c$ is a hyper-parameter, and $n_p$ is the number of visits of the parent of $X$. UCB considers both exploitation and exploration and will be used for node selection. MCTS iteratively selects a leaf node of the tree for expansion. Each iteration can be divided into four steps: selection, expansion, simulation, and back-propagation. Starting from the root node, selection is to recursively select a node with a larger UCB until a leaf node, denoted as $X$. Expansion is to execute a certain action in the state represented by $X$ and transfer to the next state, e.g., move forward and arrive at a new position in path planning. We use the child node $Y$ of $X$ to represent the next state. Simulation is to obtain

the value $v_Y$ via random sampling. Back-propagation is to update the value and the number of visits of $Y$'s ancestors.

## 3 METHOD

Hybrid CtrlFormer uses a Transformer-based Monte-Carlo tree search method to efficiently evaluate and partition the hybrid action space into good and bad subregions along the tree. In this part, we introduce Hybrid CtrlFormer in detail. To visualize the feasibility of our approach, we provide pseudo-code for the complete framework and a summary analysis of the process at the end of this section.

### 3.1 HYBRID ACTION MONTE CARLO TREE SEARCH

In the parameterized action space, each discrete action $k$ corresponds to a continuous parameter space $\mathcal{X}_k$. As the training continues, the agent has learned some knowledge about the candidate actions, thus keeping exploring the whole discrete action space and the whole continuous action space is inefficient. To reduce the number of samples needed to learn a good policy, we propose to learn to partition the overall hybrid action space into good and bad subregions (with high Q-values and low Q-values). Then, we can easily get a much smaller but promising action space by discarding the low potential regions and only keeping the high potential ones.

**High Level Idea**: As shown in Figure 2 (left), we use a hybrid action space tree to show the hierarchy of hybrid action space. The hierarchically structured action space contains $K$ discrete actions shown in blue, and each discrete action $k$ has a continuous parameter space $\mathcal{X}_k$ marked with rounded rectangles in light origin. At state $s$, ❶ for the discrete ac-

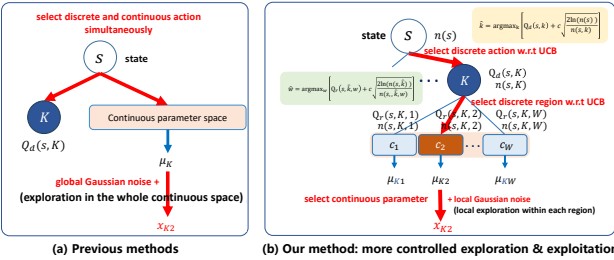

Figure 3: Difference between our method and others.

tions, we maintain a discrete Q-table $Q_d^{\pi}(s,k)$ to record their Q-values. For the simplicity of presentation, we omit the superscript $\pi$ of all Q-functions in the following. ❷ To achieve a more controlled exploration over the continuous parameter space, we split each $\mathcal{X}_k$ into $W$ subregions, i.e., $\mathcal{X}_k = \{c_{k1}, \cdots, c_{kW}\}$. We denote $\mathcal{W} = \{1, \ldots, W\}$ and also maintain a discrete Q-table $Q_r(s,k,w)$ to record the Q-value of selecting each region $w \in \mathcal{W}$ given the selected discrete action $k$. With $Q_r(s,k,w)$, we could have a general idea of how good and bad different subregions are and reduce the search space by discarding those with observably low Q-values. ❸ To provide targeted policies based on $W$ different subregions. The continuous policy of subregion $c_{kw}$ is defined as: $x_{k,w} = \mu(s; \theta_{kw})$, where $\mu$ is a deterministic policy network with parameter $\theta_{kw}$. ❹ In order to evaluate the Q-values of selecting discrete action $k$ and continuous parameter $x_{kw}$, we build a value estimation model $Q(s,k,x_{kw}; \omega)$, where $\omega$ indicates the network's parameter. ❺ For each subregion $c_{kw}$, we also maintain a set $\mathbb{X}_{kw}$ and a set $\mathbb{Q}_{kw}$ to record all previous selected parameter actions and their corresponding Q-values respectively. Based on these functions, we perform MCTS over the hybrid action tree to get the target discrete and continuous actions while balancing exploration and exploitation.

**The difference** between our method and the previous method is shown in Figure 3. previous methods have to search the whole discrete and continuous action space simultaneously according to the state which results in the explosion of exploration space dimensions. The large search space makes the policy unable to find the optimal action efficiently. However, our tree-like hierarchical action space significantly reduces the exploration space dimension (denote as $dim$), i.e., $discrete_{dim} \times continuous_{dim} \to discrete_{dim} + continuous\ subregions_{umber} + subregion_{dim}$.

### 3.1.1 The Search Procedures

Our policy searching approach is based on MCTS. Each search episode includes three stages: selection, sampling and back-propagation. This searching process is repeated for $N_{sim}$ episodes before selecting the final hybrid action. For any state $s$, $Q_d(s,k)$ and $Q_r(s,k,w)$ are initialized to zeros and $\mathbb{X}_{kw}$ and $\mathbb{Q}_{kw}$ are initialized to empty sets,

i.e., $\mathbb{X}_{kw} = \{\}$ and $\mathbb{Q}_{kw} = \{\}$, at the beginning of the search. For each search episode, we sequentially perform the following 4 steps:

**(1) Select a discrete action $\hat{k}$ w.r.t UCB.** As shown in Figure 2 (left), given the state $s$, to select a hybrid action $(\hat{k}, x_{\hat{k}\hat{w}})$ for the current search episode, we first query the Q-values for all discrete actions: $\boldsymbol{q}_d = \{Q_d(s,1), \ldots, Q_d(s,K)\}$. Then, we follow the UCB rule [Auer et al., 2002] to select a discrete action $\hat{k}$ while balancing the exploration and exploitation:

$$\hat{k} = \arg\max_{k \in \mathcal{K}} \left( Q_d(s,k) + c\sqrt{\frac{2\ln n(s)}{n(s,k)}} \right) \quad (2)$$

where $n(s)$ is the visit number of the node $s$, which is equal to the number of episodes that have been searched, $n(s,k)$ is the number of times of selecting action $k$ at node $s$ and $c$ is a constant to balance the exploration and exploitation.

**(2) Select a discrete region $\hat{w}$ w.r.t UCB.** After getting the selected discrete action $\hat{k}$, we use it to query the region Q-table $Q_r(s,k,w)$ and get the Q-values for all discrete regions: $\boldsymbol{q}_r = \{Q_r(s,\hat{k},1), \ldots, Q_r(s,\hat{k},W)\}$. We also follow the UCB rule to select a region $\hat{w}$:

$$\hat{w} = \arg\max_{w \in \mathcal{W}} \left( Q_r(s,\hat{k},w) + c\sqrt{\frac{2\ln n(s,\hat{k})}{n(s,\hat{k},w)}} \right) \quad (3)$$

By selecting a discrete action $\hat{k}$ and a discrete region $\hat{w}$ according to the previous learned knowledge (i.e., the learned Q-values and visit counts), the search space is significantly reduced and we only have to pick a continuous parameter $x_{\hat{k}\hat{w}}$ within the region $\hat{w}$ under discrete action $\hat{k}$.

**(3) Sample a continuous parameter action $x_{\hat{k}\hat{w}}$.** To finally get the parameter action, we first pick the corresponding actor $\mu(s; \theta_{\hat{k}\hat{w}})$ learned within the region $\hat{w}$ under action $\hat{k}$ and get $x_{\hat{k}\hat{w}}^0 = \mu(s; \theta_{\hat{k}\hat{w}})$. Then we sample a parameter action locally from a Gaussian distribution centered at $x_{\hat{k}\hat{w}}^0$ with a covariance matrix $\sigma^2 I$ as the final selected continuous parameter action, i.e., $x_{\hat{k}\hat{w}} \sim \mathcal{N}\left(x_{\hat{k}\hat{w}}^0, \sigma^2 I\right)$, where $\sigma$ is a small constant to control the degree of the local exploration.

**(4) Back-propagate the Q-value.** After selecting the discrete action $\hat{k}$ and the continuous parameter $x_{\hat{k}\hat{w}}$, we first compute the estimated Q-value for these newly sampled actions as $V_{new} = Q(s,\hat{k}, x_{\hat{k}\hat{w}}; \omega)$. Then, we add this newly sampled parameter action and its Q-value, to $\mathbb{X}_{kw}$ and $\mathbb{Q}_{kw}$ respectively, i.e., $\mathbb{X}_{kw}$.append($x_{\hat{k}\hat{w}}$) and $\mathbb{Q}_{kw}$.append($V_{new}$). After that, we back-propagate $V_{new}$ for updating: ❶ the visit-count and the Q-value of the selected subregion, i.e., $n(s,\hat{k},\hat{w})+ = 1$ and $Q_r(s,\hat{k},\hat{w}) = \max \mathbb{Q}_{kw}$. ❷ the visit-count and the Q-value of the selected discrete action, i.e., $n(s,\hat{k})+ = 1$ and $Q_d(s,\hat{k}) =$

$\max_w Q_r(s, \hat{k}, w)$, and ❸ the visit-count of the current state, i.e., $n(s) += 1$.

### 3.1.2 Select a Target Hybrid Action

When we finish the $N_{sim}$ search episodes, we will obtain the visit count distribution $\beta(k|s)$ of the discrete actions, i.e.,

$$\beta(k|s) = \frac{n(s, k)}{n(s)} \qquad (4)$$

and the visit count distribution $\hat{\beta}(w|s, k)$ of the discrete subregions under action $k$, i.e.,

$$\hat{\beta}(w|s, k) = \frac{n(s, k, w)}{n(s, k)} \qquad (5)$$

Then, we sample a discrete action according to $\beta(k|s)$, i.e., $\hat{k} \sim \beta(k|s)$ as the final discrete action and sample a sub-region according to $\hat{\beta}(w|s, \hat{k})$, i.e., $\hat{w} \sim \hat{\beta}(w|s, \hat{k})$ as the final subregion. Next, we select the best parameter action previously sampled in subregion $\hat{w}$ under discrete action $\hat{k}$ as the final parameter action, i.e.,

$$\hat{x}_{\hat{k}, \hat{w}} = \arg \max_{x \in \mathbb{X}_{\hat{k}, \hat{w}}} \mathbb{Q}_{\hat{k}, \hat{w}}(x) \qquad (6)$$

### 3.1.3 Better Initialization for $Q_d(s, k)$ and $Q_r(s, k, w)$

According to the original method in 3.1.1, both $Q_d(s, k)$ and $Q_r(s, k, w)$ are initialized to zeros, which do not contain much information. To give a better Q-value initialization, we jointly compute all Q-values of all discrete actions and subregions in a batch mode as:

$$Q_r(s, k, w) = Q(s, k, \mu(s; \theta_{kw}); \omega), \ \forall k \in \mathcal{K}, \forall w \in \mathcal{W} \qquad (7)$$

For each subregion $w$ under discrete action $k$, we use the corresponding continuous policy's output, i.e., $\mu(s; \theta_{kw})$, as the initialized parameter action. Then, $Q_d(s, k)$ is initialized as $Q_d(s, k) = \max_w Q_r(s, k, w)$.

### 3.2 HYBRID CTRLFORMER

The remaining question is how to efficiently implement the continuous policy networks $\mu(s; \theta_{kw}), \forall k \in \mathcal{K}, \forall w \in \mathcal{W}$ and the Q-value estimation network $Q(s, k, x_{kw}; \omega)$. The straightforward way to implement the continuous policy networks is using a set of individual MLPs for each $k$ and $w$ to perform independent evaluations. However, this will greatly increase the parameters of the framework and reduce the training efficiency [Li et al., 2021].

To reduce the parameter number while ensuring the representational ability of the dependencies of action spaces, we apply the Causal Transformer in our case to model the

dynamic relationships between different discrete actions and subregions. Transformer offers the advantage of serially processing dependencies among states, discrete actions, and continuous action subregions, akin to natural language processing [Radford and Narasimhan, 2018]. Experimental results validates the effectiveness of this design.

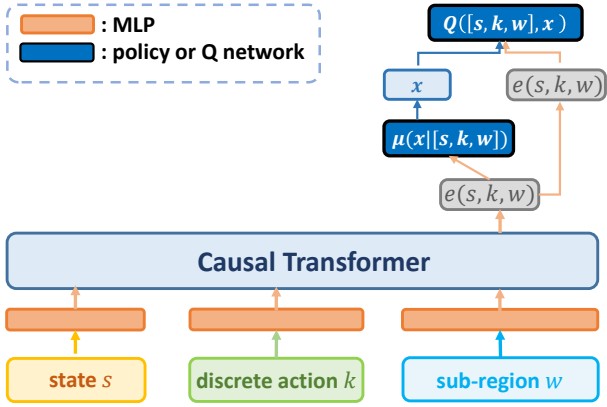

Figure 4: Hybrid Control Transformer

### 3.2.1 Architecture

The architecture of our Hybrid CtrlFormer is shown in Figure 4. The input to the Transformer contains the 3 tokens along a search episode (one for each modality): the current state $s$, one of the discrete action $k$, and one of the subregion $w$ under action $k$. To obtain token embeddings, we learn a linear layer for each modality, which projects the raw input to the embedding dimension. Both the discrete action and the subregion are converted to the one-hot format before inputting them into the model. The tokens are then processed by the Causal Transformer model, which uses causal masks to ensure that each token can only see its previous tokens.

After that, we get the Transformer's outputs, which represent the embedding vectors for the corresponding input tokens. We only pick the last embedding vector corresponding to the subregion $w$ and denote it as $e(s, k, w)$. As $e(s, k, w)$ contains all the 3 input tokens information, we can feed it into a shared policy network to get the continuous actor for each $k$ and each $w$, i.e., $\mu(s, k, w) = \mu(e(s, k, w); \theta)$, where $\theta$ is the network's parameter. We denote the output of the actor as $x_{kw}$. Next, we feed both $e(s, k, w)$ and $x_{kw}$ into a Q-network to get the estimated Q-value for the discrete action $k$ and the continuous parameter $x_{kw}$, i.e., $Q(s, k, x_{kw}) = Q(e(s, k, w), x_{kw}; \omega)$, where $\omega$ is the network's parameter.

### 3.2.2 Training the Models

The optimization loss function of $Q(e(s, k, w), x_{kw}; \omega)$ and $\mu(e(s, k, w); \theta)$ are very similar to that of DDPG [Lillicrap

et al., 2016]. We update $Q(e(s, k, w), x_{kw}; \omega)$ by minimizing the following loss:

$$\mathcal{L}^Q(\omega) = \frac{1}{2}[Q(e(s, k, w), x_{kw}) - y]^2 \qquad (8)$$

where

$$y = r + \gamma \max_{k \in \mathcal{K}, w \in \mathcal{W}} Q(e(s', k, w), \mu(e(s', k, w); \theta^-); \omega^-)$$

and $s'$ denotes the next state, $\gamma$ is the discount factor and $\theta^-$ and $\omega^-$ are the parameters of the target networks. We update the actor by minimizing the following loss:

$$\mathcal{L}^\mu(\theta) = -Q(e(s, k, w), \mu(e(s, k, w); \theta); \omega) \qquad (9)$$

---

**Algorithm 1** Hybrid Action Monte Carlo Tree Search

---

**Parameters:** $\boldsymbol{q}_d = \{Q_d(s, 1), ..., Q_d(s, k)\}$
$\quad n(s) = \{n(s, 1), ..., n(s, k)\}$
$\quad \boldsymbol{q}_r = \{Q_r(s, \hat{k}, 1), ..., Q_r(s, \hat{k}, W)\}$
$\quad n(s, K, w) = \{n(s, 1, 1), n(s, 1, 2), ..., n(s, k, w)\}$
$\quad$ The buffer that stores the history selection $\mathbb{X}_{kw}, \mathbb{Q}_{kw}$
$\quad$ Sample times per round $N_{sim}$ and exploration weight $c$
**Process:**
$\quad$ Traverse $k \times w$ continuous action subregions. Update $n(s), n(s, K, w)$.
$\quad$ estimate $\boldsymbol{q}_r$ by Transformer style critic. $\boldsymbol{q}_r(s, k, w) = Q(e(s, k, w), x_{kw}; \omega)$.
$\quad \boldsymbol{q}_d$ is initialized as $Q_d(s, k) = \max_w Q_r(s, k, w)$
$\quad$ **while** $t < N_{sim}$ **do**
$\quad\quad$ Select $\hat{k}$ by UCB $\qquad\qquad\qquad \triangleright$ Eq 2
$\quad\quad$ Select $\hat{w}$ by UCB $\qquad\qquad\qquad \triangleright$ Eq 3
$\quad\quad$ Sample continuous action by actor: $x_{\hat{k}, \hat{w}} = \mu(e(s, k, w); \theta)$
$\quad\quad V_{\text{new}} = Q(s, \hat{k}, \quad x_{\hat{k}\hat{w}}; \omega)$
$\quad\quad \mathbb{X}_{kw}.\text{append}(x_{\hat{k}\hat{w}}), \quad \mathbb{Q}_{kw}.\text{append}(V_{\text{new}})$
$\quad\quad$ Back-propagate
$\quad$ **end while**
$\quad$ Obtain $\beta(k|s)$ $\qquad\qquad\qquad\qquad \triangleright$ Eq 4
$\quad$ Get $\hat{\beta}(w|s, k)$ $\qquad\qquad\qquad\qquad \triangleright$ Eq 5
$\quad$ Sample $\hat{k}, \hat{w}$ $\qquad\qquad\qquad\qquad \triangleright$ Eq 6
$\quad$ Training the Transformer style critic $\qquad \triangleright$ Eq 8
$\quad$ Training the actor $\qquad\qquad\qquad\quad \triangleright$ Eq 9

---

### 3.3 PROCESS ANALYSIS OF HYBRID CTRLFORMER

Hybrid CtrlFormer Combine Transformer and MCTS reasonably. Firstly, MCTS is used to filter the subregion of continuous action to simplify the exploration space. Since there naturally exists hierarchical dependencies during the action

selection process [Li et al., 2021], we use a Transformer-style critic to model dependencies and generate the Q-values of each region (as Transformer is suitable for modeling sequence dependence). We regard the embedded states, actions, continuous action. Algorithm 1 shows the complete process of Hybrid CtrlFormer.

Furthermore, to prove that Hybrid CtrlFormer can solve multi-agent tasks, **Appendix C** features a theoretical proof and practical example of how Hybrid CtrlFormer can be combined with the value decomposition framework (mainstream framework of multi-agent reinforcement learning). Additionally, experimental verification of its effectiveness can be found in **Appendix D**.

## 4 EXPERIMENTS

### 4.1 EXPERIMENT SETUPS

**Benchmarks.** Figure 10 visualizes the evaluation benchmarks. Hard Goal [Hausknecht and Stone, 2016], Catch Point [Xiong et al., 2018], Move [Li et al., 2021], Platform [Masson et al., 2016], and Chase. **Hard Goal and Chase** are harder than other games. Both games require the agent to select an action from a large hybrid action space, and Chase has a complex dynamic state transition due to the target moving fast to run away. **Appendix B** contains the settings for each environment.

**Baselines.** In single-agent tasks, 5 SOTA approaches are selected as baselines: PADDPG [Hausknecht and Stone, 2016], PDQN [Xiong et al., 2018], HPPO [Fan et al., 2019], HHQN [Fu et al., 2019] and HyAR [Li et al., 2021].

**In multi-agent tasks**, five baselines are selected, Multi-agent Hybrid CtrlFormer outperforms other methods in almost MA benchmarks. Please refer to **Appendix D** for experimental results and analysis.

### 4.2 PERFORMANCE EVALUATION

To counteract the implementation bias, we directly use the official code published in their original paper. Hybrid Ctrl-Fomer samples 32 times per step. Continuous action space segmentation and hyperparameter settings can be found in **appendix E**. The summarized comparisons of the average win rates are presented in Table 1, while the learning curves of each algorithm are plotted in Figure 5 to provide a direct reflection of their learning performances. Overall, our findings indicate that Hybrid CtrlFormer achieves better results and lower variances than baselines in almost all benchmarks. In simple tasks, the experimental results of Hybrid CtrlFormer show obvious advantages over PDQN, HHQN, HPPO, and PADDPG. Moreover, our method performs better than HyAR in most scenarios. However, the simplicity of these single-agent tasks, along with their limited exploration

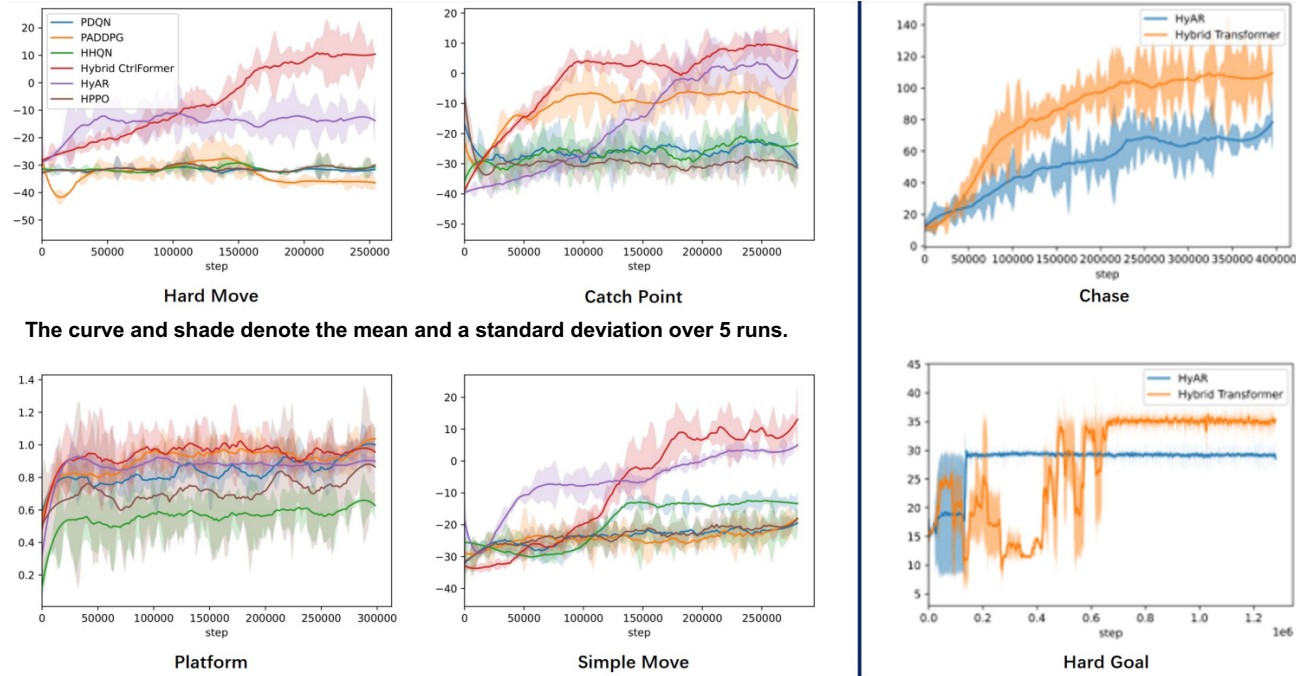

Figure 5: Left part shows the results of all methods in simple tasks. Right part compares the best baseline (HyAR-TD3) and Hybrid CtrlFormer in hard benchmarks. The $x-$ and $y-$ axis denote the environment steps and reward.

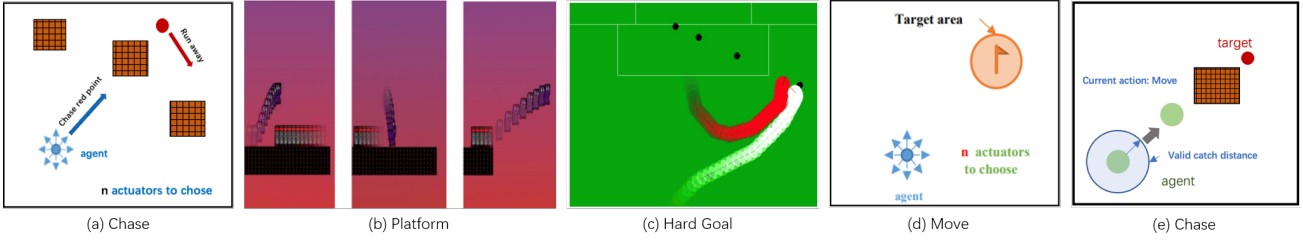

Figure 6: Hybrid action space benchmarks.

space, means that even suboptimal exploration techniques can attain high scores. For instance, while HyAR does not perform as well as our method, it also yields acceptable results across four simple tasks. To demonstrate the effective exploration of Hybrid CtrlFormer in complex hybrid action spaces, we add two more challenging tasks (Chase and Hard Goal). Our method outperforms HyAR in the final performance of both tasks. Specifically, in Hard Goal, HyAR converges faster in the early stage but eventually falls into a sub-optimal policy, while our method achieves better performance by efficiently exploring in the early stage to avoid such sub-optimal policies. To further prove the ability of our method, **four multi-agent scenarios** with local observation are added, in which Hybrid CtrlFormer has outstanding performance (Appendix D).

# 5 ABLATION STUDY

## 5.1 THE INFLUENCE OF SEGMENTING SUBREGIONS

All algorithms are tested in the same environment (i.e. MOVE) with identical hyperparameters. The continuous action space of environment is $[-1, 1]$ which is evenly divided into 1, 4 and 8 subregions. 4 sub-regions setting: [-1, -0.5), [-0.5, 0), [0,0.5), [0.5, 1]. 8 sub-regions setting: [- 1, -0.75), [-0.75, -0.5), [-0.5, -0.25), [-0.25, 0), [0,0.25), [0.25, 0.5), [0.5, 0.75), [0.75, 1.0], when it has only one sub-region, it is equivalent to removing the sub-region partitioning mechanism. The experimental results (left of figure 7) demonstrate that relying on a single, complex exploration space results in policy learning failure. However, as the density of the region division increases, the policy achieves higher scores in the same environment. This provides that the subregion partition module is effective in mitigating the convergence

| LEVEL | ENV | HPPO | PADDPG | PDQN | HHQN | HyAR | Hybrid CtrlFormer |
|-------|-----|------|--------|------|------|------|-------------------|
| Simple | Catch Point | $0.69 \pm 0.09$ | $0.82 \pm 0.06$ | $0.77 \pm 0.07$ | $0.31 \pm 0.06$ | $0.85 \pm 0.06$ | $0.98 \pm 0.03$ |
| | Platform | $0.80 \pm 0.02$ | $0.36 \pm 0.06$ | $0.93 \pm 0.05$ | $0.46 \pm 0.25$ | $0.98 \pm 0.01$ | $0.96 \pm 0.03$ |
| | Move (n=4) | $0.19 \pm 0.12$ | $0.13 \pm 0.01$ | $0.69 \pm 0.07$ | $0.39 \pm 0.14$ | $0.88 \pm 0.04$ | $0.94 \pm 0.02$ |
| | Move (n=8) | $0.14 \pm 0.11$ | $0.16 \pm 0.06$ | $0.14 \pm 0.08$ | $0.09 \pm 0.08$ | $0.89 \pm 0.09$ | $0.97 \pm 0.04$ |
| Hard | Chase (n=8) | $0.12 \pm 0.03$ | $0.08 \pm 0.01$ | $0.14 \pm 0.12$ | $0.13 \pm 0.02$ | $0.43 \pm 0.13$ | $0.81 \pm 0.08$ |
| | Hard Goal | $0.08 \pm 0.06$ | $0.17 \pm 0.04$ | $0.07 \pm 0.03$ | $0.15 \pm 0.03$ | $0.30 \pm 0.18$ | $0.58 \pm 0.13$ |

Table 1: Comparisons of the baselines regarding the average win rate at the end of the training process with the corresponding standard deviation over 5 runs. Lightgreen indicates the second-best results and darkgreen denotes the best performances.

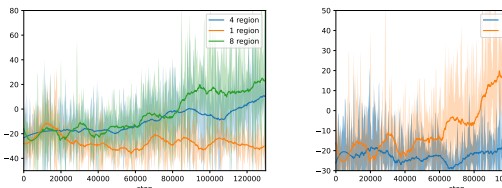

Figure 7: Learning curves of two ablation studies. The left part methods are all Transformer based. The curve and shade denote the mean and a standard deviation over 5 runs.

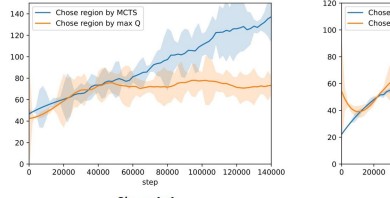
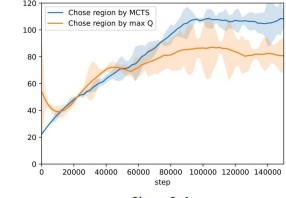

Figure 8: Necessity study of MCTS. The curve and shade denote the mean and a standard deviation over 5 runs.

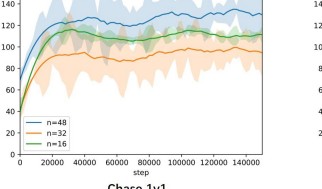
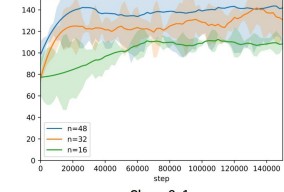

Figure 9: Effect of sampling steps on MCTS. The curve and shade denote the mean and a standard deviation over 5 runs.

difficulties caused by the complexity of the action space.

## 5.2 EFFECTIVENESS ANALYSIS OF TRANSFORMER

We investigated whether a Transformer-style sequence model can effectively capture hybrid action space dependencies and generate accurate Q-values. Thus, we replace the Transformer-based Q-network with a standard MLP (consistent with the approach taken in HPPO) while ensuring that the network depth remained constant. The results (right part of Figure 7) demonstrate that the Transformer-style model outperforms the MLP in capturing action space dependencies, thus validating its effectiveness in this context. Besides, the model computational efficiency analysis in **Appendix G** proves that Hybrid CtrlFormer does not make the model redundant although Transformer is employed.

## 5.3 WHETHER MCTS IS NECESSARY

To demonstrate the importance of MCTS in evaluating dense subregions, Hybrid CtrlFormer is reduced to a rigid rule as the baseline (i.e., regions with the highest Q-values for each sampling), we conducted experiments in two scenarios. The results, as presented in Figure 8, show that the MCTS-based approach significantly outperforms the baseline approach. This finding underscores the effectiveness of MCTS in balancing exploration and exploitation, thereby enhancing the efficiency of the learning process.

## 5.4 THE EFFECT OF MCTS SAMPLING TIMES

To investigate the impact of sampling steps on MCTS, we conducted experiments in two distinct scenarios (Figure 9). While keeping all other modules constant, we varied the sampling times to 16, 32, and 48, respectively. Although both $n = 16$ and $n = 32$ have advantages and disadvantages in different environments, they are both lower than $n = 48$. Our findings indicate that a higher number of sampling steps (e.g., 48) can enable MCTS to more accurately evaluate each action subregion based on current state information.

## 6 CONCLUSION

This paper proposes Hybrid CtrlFormer, a Transformer based deep RL method that can achieve efficient exploration in hybrid action space. Hybrid CtrlFormer sequential partitions the continuous action space into several subregions, then use a Transformer-based Monte-Carlo tree search method to efficiently evaluate and partition the hy-

brid action space into good and bad regions along the tree. Finally, Our experiments demonstrate the superiority of Hybrid CtrlFormer regarding performance, efficiency of exploration and robustness in most single-agent and multi-agent hybrid action environments, especially in complex tasks.

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

# Appendix

**Jiashun Liu**[*1]    **Xiaotian Hao**[*1]    **Jianye Hao**[†1]    **Yan Zheng**[1]    **Yujing Hu**[2]    **Changjie Fan**[2]    **Tangjie Lv**[2]

**Zhipeng Hu**[2]

[1]Tianjin University, Tianjin, China
[2]FUXI AI Laboratory, NetEase, Hangzhou, China

## A RELATED WORKS

Three types of methods have been proposed for Parameterized Markov Decision Processes(PAMDP). The methods of the first type convert the heterogeneous action space into a homogeneous one by either discretizing continuous action space ([Massaroli et al., 2020]) or transferring discrete actions into continuous space[Hausknecht and Stone, 2016]. By doing this, conventional RL algorithms can be straightly applied in this setting. However, discretizing all dimensions of the continuous action suffers from the loss of control accuracy and the scalability issue (due to the exponentially exploring number of discretized actions); while casting all discrete actions into continuous ones enlarges the original action space, resulting in additional difficulties in approximation and generalization[Li et al., 2021].

In order to avoid the problems caused by direct homogeneous motion space. The methods of the second type directly build separate policies for the discrete actions and the continuous ones. Hybrid PPO (HPPO) [Fan et al., 2019] builds multiple network heads (one for discrete actions and the others for the continuous parameters of each discrete action) to learn the hybrid policy. A similar idea is also adopted in Action Branching PPO. However, building a separate continuous policy network for each discrete action will significantly enlarge the model size, resulting in additional difficulties in optimization. Besides, both these two types of methods neglect the natural dependence between each discrete action and the corresponding continuous parameters, thus their learning processes are inefficient especially when the hybrid action space becomes high-dimensional.

To improve the sample efficiency, The methods of the third type explicitly incorporate such dependencies into the model design. Parameterized DQN (PDQN) [Xiong et al., 2018] proposes a hybrid structure by combining DQN [Silver et al., 2016] with DDPG [Lillicrap et al., 2016], where the Q-network of DQN directly takes the actor's outputs of DDPG (i.e., the continuous parameters for all discrete actions) as additional inputs. However, the dependence of PDQN's Q-values on all parameters actions causes a *false gradients* issue and can lead to suboptimal action selection. To address the *false gradients* issue. *False gradients* means that due to the serial training of discrete action strategies and all continuous action strategies, the gradient of discrete action policy affects the training of unrelated continuous action modules, making the overall strategy suboptimal. Bester et al. [2019] further design a Multi-Pass Q-Network (MP-DQN), making each discrete action's Q-value only depends on its corresponding continuous parameters. Similar ideas also have been applied to Soft Actor Critic (SAC) based methods [Delalleau et al., 2019] and PPO based methods [Ma et al., 2021]. And [Fu et al., 2019] further extends PDQN to multi-agent RL settings. Most recently, Li et al. [2021] propose Hybrid Action Representation (HyAR), which uses an embedding table and a conditional Variational Auto-Encoder (VAE) to convert both the discrete and the continuous actions into a more compact latent space. Then, the policy is trained in the latent action space via Twin Delayed Deep Deterministic policy gradient (TD3) algorithm [Fujimoto et al., 2018]. HyAR achieves state-of-the-art (SOTA) performance on typical hybrid control tasks with higher sample efficiency, especially for high-dimensional action spaces. However, the embedding table and the VAE have to be periodically re-trained with the policy updating. Thus, HyAR is unstable and difficult to train due to the co-evolution of the latent action embedding and the RL policy. **Apart from the action-dependency modeling, another challenge affecting learning efficiency is the hard exploration problem**. As far as we know, this problem has not been discussed in the previous work, and our work is devoted to solving this problem so as to further improve the learning efficiency of hybrid action space control tasks.

*Accepted for the 40th Conference on Uncertainty in Artificial Intelligence* (UAI 2024).

## B EXPERIMENTAL DETAILS

Our codes are implemented with Python 3.7 and Torch 1.7.1 experiments were run on a single NVIDIA GeForce GTX 2080Ti GPU. Each single training trial ranges from 4 hours to 10 hours, depending on the algorithms and environments.

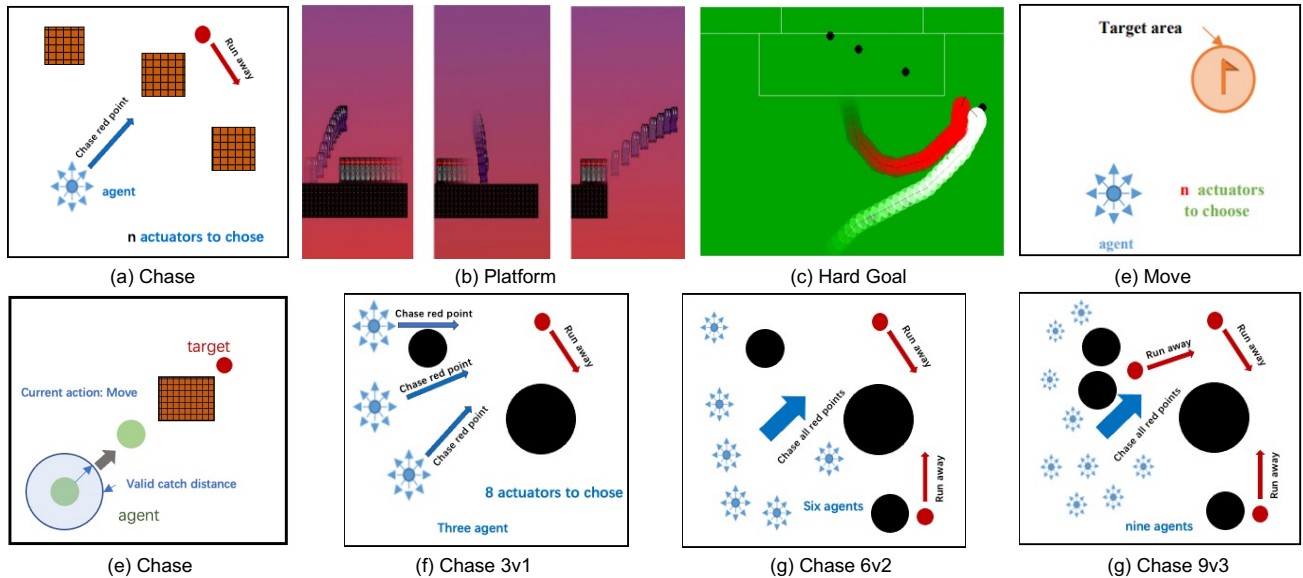

Figure 10: Benchmarks

**The multi-agent hybrid control tasks** (Figure 10 (f) 3v1,(g) 9v3, require cooperation among agents to achieve goals. The action space of the agent is the same as single agent Chase, but multiple agents need to cooperate to catch all the targets. And need to bypass roadblocks. Both environments are divided into easy and hard tasks, with the harder tasks having targets that move faster and are smaller and harder to capture.

**Single agent Environments.** We conduct our experiments on several hybrid action environments and a detailed experiment description is below.

- Platform: The agent needs to reach the final goal while avoiding the enemy or falling into the gap. The agent needs to select the discrete action (run, hop, leap) and determine the corresponding continuous action (horizontal displacement) simultaneously to complete the task. The horizon of an episode is 20.

- Hard Goal: Three types of hybrid actions are available to the agent including kick-to $(x, y)$, shoot-goal $(h)$. The shot-goal action and split into ten parameterized actions by dividing the goal line equidistantly. The continuous action parameters of each shot action will be mapped to a region in the goal line. The horizon of an episode is 50.

- Catch Point: The agent should catch the target point (orange) in limited opportunity (10 chances). There are two hybrid actions move and catch. Move is parameterized by a continuous action value which is a directional variable and catch is to try to catch the target point. The horizon of an episode is 20.

- Chase (designed by us): The agent needs to control n equally spaced actuators to reach the target area (orange). The agent can choose whether each actuator should be on or off. Thus, the size of the action set is exponential in the number of actuators that is $2^n$. Each actuator controls the moving distance in its own direction. n controls the scale of the action space. As n increases, the dimension of the action will increase. The horizon of an episode is 25.

- Multi-agent Chase (designed by us): The action space of the agent is the same as Chase, but multiple agents need to cooperate to catch all the targets. And need to bypass roadblocks.

## C MULTI-AGENT HYBRID CTRLFORMER

Hybrid CtrlFormer is an actor-critic framework and can be easily combined with existing value-based or actor-critic based multi-agent algorithms. The most straightforward way to introduce Hybrid CtrlFormer into multi-agent RL is using

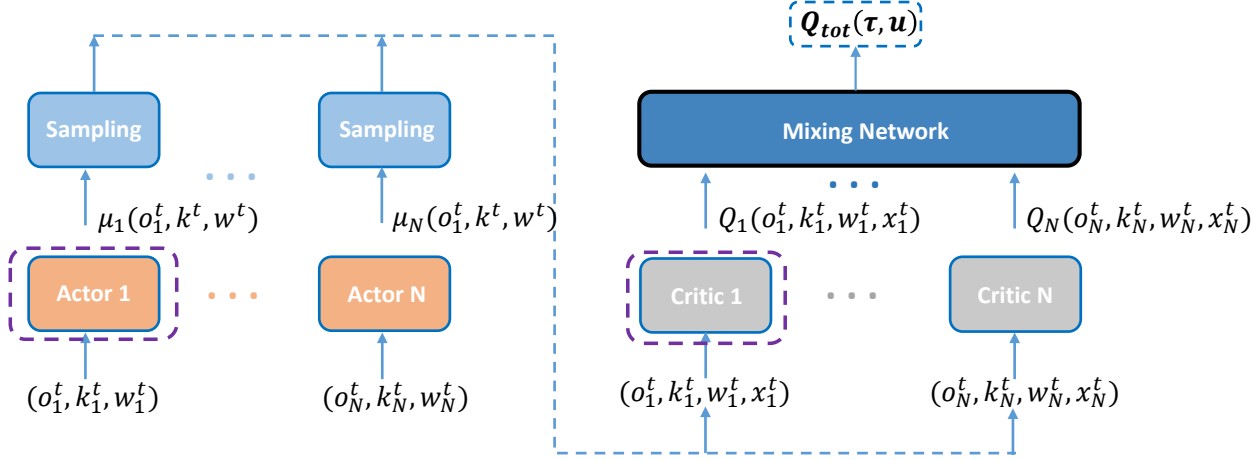

Figure 11: Multi-agent Hybrid CtrlFormer

independent Q-learning (IQL) [Tan, 1993]. However, this simple approach suffers from the non-stationarity issue caused by the changing policies of the learning agents, and thus the learning process is unstable. Recently, a great deal of work has shown that using centralised training can naturally handle the non-stationary problem [Yang et al., 2020, Wang et al., 2020]. Inspired by FACMAC [Peng et al., 2021], we utilize the value decomposition mechanism to coordinate the policy update over hybrid action spaces among agents.

The overall architecture of MA Hybrid CtrlFormer is shown in Figure 11. In MA-Hybrid CtrlFormer, each agent's continuous policy is $\mu_i(e(s, k, w); \theta)$ and each agent's Q-network is $Q_i(e(s, k, w), x_{kw}; \omega)$ and all agents share the same parameters ($\theta$ and $\omega$) to facilitate training. All agents share a centralised critic $Q_{tot}$ which can be factored as:

$$Q_{tot}(\boldsymbol{\tau}, \boldsymbol{u}, \boldsymbol{s}; \psi) = h_\psi(\boldsymbol{s}, \{Q_i(e(s, k, w), x_{kw}), i \in \{1, \ldots, N\}\}) \tag{10}$$

where $\boldsymbol{\tau}$ denotes local action observation histories and $\boldsymbol{s}$ means the global state, $\boldsymbol{u}$ denotes all agents' hybrid actions, $\{Q_i(e(s, k, w), x_{kw}), i \in \{1, \ldots, N\}\}$ are the individual Q-values of all agents (N is the agent number), and $h_\psi$ is a non-linear monotonic function parameterized as a mixing network with parameters $\psi$, as in QMIX [Rashid et al., 2018]. The centralised but factored critic is trained by minimizing the following loss:

$$L(\psi; \omega) = \mathbb{E}_D\left[\left(y^{tot} - Q_{tot}(\boldsymbol{\tau}, \boldsymbol{u}, \boldsymbol{s}; \psi)^2\right)\right] \tag{11}$$

where $y^{tot} = r + \gamma Q_{tot}(\boldsymbol{\tau'}, \boldsymbol{\mu'}, \boldsymbol{s'}, \psi^-))$, and $\boldsymbol{\mu'}$ represents the target hybrid actions generated by all agents' target individual Q-networks and target individual actors. The individual continuous policy is optimized according to the following loss:

$$\mathcal{L}^\mu(\theta) = -Q_{tot}(\boldsymbol{s}, \{Q_i(e(s, k, w), \mu(e(s, k, w); \theta); \omega), \ldots\}) \tag{12}$$

# D   MULTI-AGENT EXPERIMENTS

In multi-agent tasks, five SOTA approaches are selected as baselines: Independent PDQN [Xiong et al., 2018], MAPDQN [Fu et al., 2019], Independent HPPO [Fan et al., 2019], Independent HyAR [Li et al., 2021] and CTDE-HyAR [Li et al., 2021]. Hybrid CtrlFormer contains one layer transformer( with one MLP, and a unique attention mechanism). Thus the network depth of algorithms is the same. PDQN, HHQN, HyAR, and Hybrid CtrlFormer are all implemented based on the same architecture DDPG. CTDE-HyAR is based on FACMAC [Peng et al., 2021].

The experimental results are shown in Figure 12, in which the total average reward curves are smoothed for visual clarity. The results show that MA Hybrid CtrlFormer, denoted as MAHT in the figure, significantly outperforms all baselines in all six scenarios. Cooperative tasks cause the environment to be unstable [Rashid et al., 2018] (the environment can be changed by the actions of other agents), which makes the state information more complex. This phenomenon reinforces the need for exploration efficiency. Moreover, the performance of our method is stable in the large-scale agent scenarios, which proves that MA Hybrid CtrlFormer can not only realize effective exploration but also overcome the factors of environmental instability to achieve efficient coordination between agents by combining the value decomposition method.

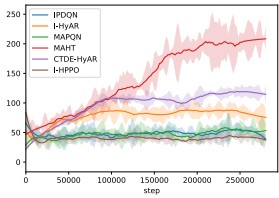 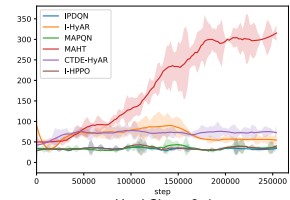 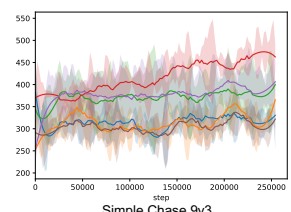 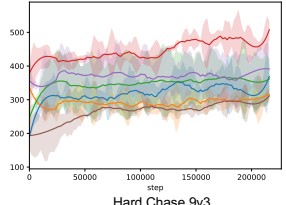

| | | | | | | |
|---|---|---|---|---|---|---|
| Simple Chase 3v1 | | Hard Chase 3v1 | | Simple Chase 9v3 | | Hard Chase 9v3 |

Figure 12: Comparisons of algorithms in multi-agent environments. $y-$axis denotes the average reward. The curve and shade denote the mean and a standard deviation over 5 runs. *We abbreviate Multi-agent Hybrid CtrlFomer to MAHT.*

| Hyperparameter | HPPO | PADDPG | PDQN | HHQN | HyAR-DDPG | Hybrid CtrlFormer |
|---|---|---|---|---|---|---|
| Actor Learning Rate | $1e^{-4}$ | $1e^{-4}$ | $1e^{-4}$ | $1e^{-4}$ | $1e^{-4}$ | $1e^{-4}$ |
| Critic Learning Rate | $1e^{-3}$ | $1e^{-3}$ | $1e^{-3}$ | $3e^{-4}$ | $3e^{-4}$ | $1e^{-4}$ |
| Target Actor Update Rate | $N$ | $1e^{-3}$ | $1e^{-3}$ | $1e^{-3}$ | $1e^{-3}$ | $5e^{-3}$ |
| Target Critic Update Rate | $N$ | $1e^{-2}$ | $1e^{-2}$ | $1e^{-2}$ | $1e^{-2}$ | $5e^{-3}$ |
| Batch Size | 128 | 128 | 128 | 128 | 128 | 128 |
| Buffer Size | $1e5$ | $1e5$ | $1e5$ | $1e5$ | $1e5$ | $1e5$ |

Table 2: Comparisons of the baselines regarding the average win rate at the end of the training process with the corresponding standard deviation over 5 runs. Values in bold indicate the second-best results in each environment. And Red Values denote the best performances.

# E  HYPERPARAMETERS

For all of our experiments, we use the raw state and reward from the environment without any normalization or scaling. Additionally, no regularization is applied to the actor and critic models in any of the algorithms. To encourage exploration, an exploration noise sampled from $N(0, 0.1)$ is added to all baseline methods when selecting actions. In the case of Hybrid CtrlFormer, the continuous action space of each environment is uniformly divided into 8 sub-regions in sequential order. Concretely, the continuous action space of the environment is $[-1, 1]$, and the 8 sub-regions are defined as follows: [-1, -0.75), [-0.75, -0.5), [-0.5, -0.25), [-0.25, 0), [0, 0.25), [0.25, 0.5), [0.5, 0.75), and [0.75, 1.0). The $c$ parameter of the MCTS is $1/\sqrt{2.0}$, which is the optimal setting that has been tested extensively and is the general default setting for MCTS. The number of single agent training episodes is 6000 to ensure that all algorithms converge. The episode number for MARL frameworks is 10000. The discount factor used is 0.99, and we employ the Adam optimizer for all algorithms. Table 2 summarizes the common hyperparameters used in all of our experiments.

# F  NETWORK STRUCTURE

PADDPG and PDQN are implemented with reference to `https://github.com/cycraig/MP-DQN`. For a fair comparison, all the baseline methods have the same network structure (except for the specific components of each algorithm) as our HyAR-TD3 implementation. For PDQN, PADDPG, HPPO paper does not provide open-source code and thus we implemented it by ourselves according to the guidance provided in their paper. For HPPO, the discrete actor and continuous actor do not share parameters (better than sharing parameters in our experiments). We use a two-layer feed-forward neural network of 256 and 256 hidden units with ReLU activation (except for the output layer) for the actor network for all algorithms. For PADDPG, PDQN and HHQN, the critic denotes the Q-network. For HPPO, the critic denotes the V network. Some algorithms (PADDPG, HHQN) output two heads at the last layer of the actor network, one for discrete action and another for continuous action parameters. For HyAR-DDPG, we use the author's open-source code. In order to prove that the Hybrid CtrlFormer framework is not redundant and will not cause training difficulties due to the division of continuous motion space, the structure details are shown in Tab 3.

| Model Component | Layer(Name) | Structure |
|---|---|---|
| Critic | Fully Connected | $(state\ dim, 128)$ |
| | Activation | ReLU |
| | Fully Connected | $(128, 1)$ |
| Transformer | Fully Connected | $(state\ dim, 128)$ |
| | Activation | ReLU |
| | self-attention | $(128, 128)$, ReLU |
| | Fully Connected | $(128, subregion\ number)$ |
| Actor | Fully Connected | $(128, 1)$ |

Table 3: Network structures for the Hybrid CtrlFormer including, the discrete action Q network, one layer transformer, and the actor-critic architecture

| Benchmarks | HPPO | PADDPG | PDQN | HHQN | HyAR-DDPG | Hybrid CtrlFormer |
|---|---|---|---|---|---|---|
| Platform | $2h4m57s$ | $2h24m33s$ | $2h56m06s$ | $2h26m17s$ | $3h11m41s$ | $2h48m51s$ |
| Chase | $20h34m08s$ | $20h28m14s$ | $22h06m36s$ | $22h27m12s$ | $23h20m06s$ | $22h12m35s$ |
| Goal | $7h12m23s$ | $7h15m57s$ | $7h46m05s$ | $8h03m02s$ | $8h53m26s$ | $8h41m29s$ |

Table 4: The convergence rate of Hybrid CtrlFormer is in the middle among all methods.

## G   MODEL SCALE AND PERFORMANCE ANALYSIS

Despite utilizing both Transformer and MCTS, Hybrid CtrlFormer does not increase the network depth, and the computational demand remains at the same order of magnitude as previous methods. We conducted experiments to test the convergence time of all methods (with Hybrid CtrlFormer sampling 32 times per step) in multiple scenarios on a single NVIDIA GeForce GTX 2080Ti GPU (see Tab 4). The results demonstrate that the combination of multiple modules has a limited effect on the time performance of our method, and its convergence speed remains at the same order of magnitude as previous work.

