# OpenReview forum: "Hybrid CtrlFormer: Learning Adaptive Search Space Partition for Hybrid Action Control via Transformer-based Monte Carlo Tree Search"
_auai.org/UAI/2024/Conference — UAI 2024 poster_

### Official Review · Reviewer_xieR · 2024-03-21

**Q2-1 Originality-Novelty:** 3
**Q2-2 Correctness-Technical Quality:** 3
**Q2-5 Clarity Of Writing:** 3

**Q1 Summary And Contributions:**

This study presents the Hybrid Control Transformer (Hybrid CtrlFormer), an innovative methodology aimed at achieving an optimal balance between exploration and exploitation in environments characterized by mixed discrete-continuous action spaces. The framework models the action space as a hierarchical tree, where discrete actions are placed at a higher level, and continuous actions are allocated at a lower level. To refine the search space, the continuous action domain is divided into several sub-regions. Monte Carlo Tree Search (MCTS) is utilized for the selection of discrete actions and sub-regions, each associated with a distinct Q-table to facilitate unique evaluations. The Upper Confidence Bound (UCB) algorithm is applied iteratively to select these elements, promoting an effective equilibrium between exploration and exploitation. A continuous action is sampled from the designated sub-region for Q-value estimation and backward propagation. Over multiple iterations, distributions for discrete actions and sub-regions are derived from visit counts and steps.

**Q2-3 Extent To Which Claims Are Supported By Evidence:**

3: Good: the main claims are supported by convincing evidence (in the form of adequate experimental evaluation, proofs, (pseudo-)code, references, assumptions).

**Q2-4 Reproducibility:**

3: Good: key resources (e.g. proofs, code, data) are available and key details (e.g. proofs, experimental setup) are sufficiently well-described for competent researchers to confidently reproduce the main results.

**Q3 Main Strengths:**

To capture the dynamics between current observations, discrete actions, and sub-regions, a causal transformer is employed. The latent features encoded by this transformer are fed into both an actor neural network and a critic neural network, facilitating the generation of continuous actions and the estimation of Q-values, respectively. The comprehensive model, encompassing the causal transformer, actor, and critic networks, undergoes updates akin to the Deep Deterministic Policy Gradient (DDPG) algorithm.

To rigorously evaluate the proposed Hybrid Control Transformer (Hybrid CtrlFormer), the authors conducted a series of experiments across four distinct environments, benchmarking their method against several state-of-the-art algorithms. Additionally, the research design included targeted experiments to dissect and understand the individual contributions of each component—namely, sub-regions, transformers, and Monte Carlo Tree Search (MCTS)—within their methodology. Furthermore, the authors extended their evaluation to multi-agent environments, aiming to assess the adaptability and effectiveness of their method in more complex scenarios.

The methodology introduced in this study exhibits considerable promise. By reducing the search space through the strategic use of discrete actions and sub-regions, and employing the Upper Confidence Bound (UCB) algorithm to finely balance exploration and exploitation, this approach demonstrates validity, as corroborated by the empirical results. Furthermore, the transformer architecture offers an efficient means of delineating and encoding the interrelations among states, discrete actions, and sub-regions, thereby facilitating the generation of continuous actions and the estimation of Q-values. The experimental design is both comprehensive and robust, further solidifying the method's credibility.

**Q4 Main Weakness:**

Couldn't identify main weaknesses.

**Q5 Detailed Comments To The Authors:**

This study presents the Hybrid Control Transformer (Hybrid CtrlFormer), an innovative methodology aimed at achieving an optimal balance between exploration and exploitation in environments characterized by mixed discrete-continuous action spaces. The framework models the action space as a hierarchical tree, where discrete actions are placed at a higher level, and continuous actions are allocated at a lower level. To refine the search space, the continuous action domain is divided into several sub-regions. Monte Carlo Tree Search (MCTS) is utilized for the selection of discrete actions and sub-regions, each associated with a distinct Q-table to facilitate unique evaluations. The Upper Confidence Bound (UCB) algorithm is applied iteratively to select these elements, promoting an effective equilibrium between exploration and exploitation. A continuous action is sampled from the designated sub-region for Q-value estimation and backward propagation. Over multiple iterations, distributions for discrete actions and sub-regions are derived from visit counts and steps.

To capture the dynamics between current observations, discrete actions, and sub-regions, a causal transformer is employed. The latent features encoded by this transformer are fed into both an actor neural network and a critic neural network, facilitating the generation of continuous actions and the estimation of Q-values, respectively. The comprehensive model, encompassing the causal transformer, actor, and critic networks, undergoes updates akin to the Deep Deterministic Policy Gradient (DDPG) algorithm.

To rigorously evaluate the proposed Hybrid Control Transformer (Hybrid CtrlFormer), the authors conducted a series of experiments across four distinct environments, benchmarking their method against several state-of-the-art algorithms. Additionally, the research design included targeted experiments to dissect and understand the individual contributions of each component—namely, sub-regions, transformers, and Monte Carlo Tree Search (MCTS)—within their methodology. Furthermore, the authors extended their evaluation to multi-agent environments, aiming to assess the adaptability and effectiveness of their method in more complex scenarios.

The methodology introduced in this study exhibits considerable promise. By reducing the search space through the strategic use of discrete actions and sub-regions, and employing the Upper Confidence Bound (UCB) algorithm to finely balance exploration and exploitation, this approach demonstrates validity, as corroborated by the empirical results. Furthermore, the transformer architecture offers an efficient means of delineating and encoding the interrelations among states, discrete actions, and sub-regions, thereby facilitating the generation of continuous actions and the estimation of Q-values. The experimental design is both comprehensive and robust, further solidifying the method's credibility.

**Q9 Complying With Reviewing Instructions:**

Yes

---

> ### Author Rebuttal · Authors · 2024-04-07
>
> We sincerely thank you for your meticulous assessment and valuable insights regarding our paper. We appreciate you recognizing the contributions of our work.
>
> We will continue to improve our submission.

---

### Official Review · Reviewer_iF2n · 2024-03-23

**Q2-1 Originality-Novelty:** 3
**Q2-2 Correctness-Technical Quality:** 3
**Q2-5 Clarity Of Writing:** 3

**Q10 Ethical Concerns:**

No.

**Q1 Summary And Contributions:**

Hybrid action control problems which control discrete actions and continuous actions simultaneously remain a challenge for its too-large exploration space. This paper develops a new algorithm Hybrid CtrlFormer which combines Transformer and MCTS. By splitting the action choices into 3 different stages, they succeed in reducing the exploration space dimension effectively. Also through simulation, they prove their method improves the learning efficiency and performance effectively.

**Q2-3 Extent To Which Claims Are Supported By Evidence:**

3: Good: the main claims are supported by convincing evidence (in the form of adequate experimental evaluation, proofs, (pseudo-)code, references, assumptions).

**Q2-4 Reproducibility:**

2: Fair: key resources (e.g. proofs, code, data) are unavailable but key details (e.g. proof sketches, experimental setup) are sufficiently well-described for an expert to confidently reproduce the main results.

**Q3 Main Strengths:**

1. The Hierarchy Method reduce the searching space effectively in a reasonable way.

2. The results are well evaluated by the simulations, especially for the ablation experiments.

**Q4 Main Weakness:**

1. The illustration for the methodology details is not that clear to me sometimes.

2. The insight behind choices in the methodology is sometimes unclear.

**Q5 Detailed Comments To The Authors:**

In general, it is a quite good paper with reasonable methodology and persuasive experiments. Not being a researcher in this specific direction, I still get some questions and suggestions:

1. The necessity and the reason why using transformer? Of course, transformer is always a fashion choice in DL, but I will still wonder if there is a better choice.

2. For you to split the control process into 3 levels, especially the sub-area choice level makes me confused even though you have proved it through ablation experiments. I would wonder about the insight behind this. Or could this problem be solved by other optimization methods like bandit in continuous space?

**Q9 Complying With Reviewing Instructions:**

Yes

---

> ### Author Rebuttal · Authors · 2024-04-07
>
> We are immensely grateful to you for recognizing the importance and novelty of our approach. Your valuable suggestions inspire us to improve our work further.
> ### Detailed Comments
> **C1: The necessity and the reason why using transformer? Of course, transformer is always a fashion choice in DL, but I will still wonder if there is a better choice.**
>
> We will highlight this part in the new version. Since there naturally exists hierarchical dependencies during the action selection process, i.e., state$\to$discrete action$\to$discrete region$\to$continuous action, we use a Transformer-style sequence model to model such dependencies and generate their Q-values (as the Transformer is suitable for modeling sequence dependence). We regard the embedded states, actions, regions, etc, as input tokens (like words in NLP). Besides, Qi,et al.[1] demonstrate that transformer has a better ability to capture relevance than MLP.
>
> **C2: For you to split the control process into 3 levels, especially the sub-area choice level makes me confused even though you have proved it through ablation experiments. I would wonder about the insight behind this. Or could this problem be solved by other optimization methods like bandit in continuous space?**
>
> Your advice is valuable for our research. We will improve this part and add additional experiments in the new version.
> - Since previous work has shown that effective policies for hybrid action control need to learn dependencies between discrete action and continuous action [2], we design the hierarchical action structure to explicitly model the correlations between discrete actions and continuous actions.
> - Such a hierarchical decision is a flow from coarse-grained (discrete actions) to fine-grained (continuous action sub-area). That is, in our hierarchical architecture, when the algorithm finds the optimal subregion, it only needs to repeat sampling in this interval without going back to the upper level to explore other subintervals, ultimately improving the exploration efficiency.
> -  Compared with the straightforward bandit method (i.e. flattening the hybrid action space, partitioning the sub-interval and then selecting the action), our method can simplify the exploration space by using the dependency between discrete actions and continuous actions, that is, directly abandoning the exploration of the continuous action space corresponding to irrelevant discrete actions.
>
> We compare our method and the Discounted UCB bandit method in three scenarios. We split the same subregions, i.e. 8, for each continuous action space for both methods to ensure a fair comparison. Table 1 shows that the exploration effect of our method is better.
>
> Table 1. Performance (Average of 10 runs):
> | Method      | Chase | Move| Hard Goal|
> | :-----------: | :-----------: | :------------: | :-----------: |
> | Ours|$0.81\pm 0.08$|$0.97\pm 0.04$|$0.58\pm 0.13$|
> | Bandit for continous space |$0.73\pm 0.11$|$0.86\pm 0.07$|$0.27\pm 0.13$|
>
>
> [1]: Han, Qi, et al.``On the connection between local attention and dynamic depth-wise convolution." ICLR (2021)
>
> [2]: Li, Boyan, et al. "Hyar: Addressing discrete-continuous action reinforcement learning via hybrid action representation." ICLR (2022)

---

### Official Review · Reviewer_pZJ4 · 2024-03-24

**Q2-1 Originality-Novelty:** 1
**Q2-2 Correctness-Technical Quality:** 2
**Q2-5 Clarity Of Writing:** 2

**Q1 Summary And Contributions:**

The paper presents a method for solving control problems where controls are defined via a set of symbolic actions with continuous arguments/parameters. The authors refer to these problems as hybrid-action control tasks and propose a method based on partitioning the continuous parameter space of individual actions into K sub-regions. The method essentially performs Monte Carlo tree search (MCTS) with a selection step involving selecting first the high-level symbolic action and then the corresponding sub-region. In this case, the MCTS uses Q-value- and policy- networks, which are implemented using transformers. The method is experimentally evaluated and showed to outperform other related methods in several 2D environments (continuous navigation-like). The method is also explored in a multi-agent setting.

**Q2-3 Extent To Which Claims Are Supported By Evidence:**

2: Fair: the main claims are somewhat supported by evidence (but the experimental evaluation may be weak, or does not match entirely with the claims, important baselines may be missing, proofs contain important ideas but lack rigor, algorithmic details are only discussed superficially, references are imprecise, assumptions are not sufficiently motivated or explicated, etc.).

**Q2-4 Reproducibility:**

2: Fair: key resources (e.g. proofs, code, data) are unavailable but key details (e.g. proof sketches, experimental setup) are sufficiently well-described for an expert to confidently reproduce the main results.

**Q3 Main Strengths:**

- The paper considers an interesting and hard problem
- The presentation of the method is clear
- The experimental analysis, including the performance evaluation and the ablation study, is interesting

**Q4 Main Weakness:**

- The contribution is rather modest and simple
- Unclear how the method contributes to better exploration in large-scale problems with hybrid action spaces
- Lacks relation with existing work dealing with hybrid action spaces, e.g., task and motion planning

**Q5 Detailed Comments To The Authors:**

The paper is generally clear and well written and presents a very simple idea that seems to work well in practice.
I have several concerns with this work:
- The first one is that it is unclear how the proposed method helps in the so-called hard exploration tasks. Algorithm 1 is confusing, requiring traversing all sub-regions first, then doing MCTS and eventually, training the transformer. This suggests that a very good initialization of the value and policy networks is needed, with samples covering most of the interesting parts of the state-action space. The examples used in the benchmarks are also small scale: a 2D continuous environment with a few obstacles, suggesting poor scalability. What about a hybrid task where, instead of reaching a goal avoiding some obstacles, the agent would need to look for a key and open a door on the other side of the map. This deserves clarification.
- Second, the proposed methodology appears too incremental. The authors propose a strategy to encode/represent continuous parameterized actions which can be efficient for MCTS but, unfortunately, there is no formal analysis besides the empirical evaluation. For example, why an homogeneous partitioning? It is unclear how this strategy would extend to other hybrid settings beyond the ones used in the experiments.
- Third, the paper claims to address several challenges: hybrid action spaces, hard (sparse) exploration problems, and eventually mult-agent systems. It is confusing where is the main contribution of the work.
- Finally, there is a vast literature combining discrete (for example combined task and motion planning) that is not mentioned at all in this work.

**Q9 Complying With Reviewing Instructions:**

Yes

---

> ### Author Rebuttal · Authors · 2024-04-07
>
> ## Part 1
> Thanks for your thorough, meticulous, and impartial advice which considerably eases the enhancement of our research. Your valuable suggestions inspire us to improve our work further. If you think the following response addresses your concerns, we would appreciate it if you could kindly consider raising the score.
>
>
> ### Questions
> **Q1.1: The examples used in the benchmarks are also small scale: a 2D continuous environment with a few obstacles, suggesting poor scalability. What about a hybrid task where, instead of reaching a goal avoiding some obstacles, the agent would need to look for a key and open a door on the other side of the map. This deserves clarification.**
>
> Thanks for your valuable advice! We will add validation on complex scenarios in the new version. We use minigrid to construct a set of maze tasks with the same goal: The agent must take the key to open the door and go to the goal point. These three scenes have the same three targets, but different map sizes and boundary ranges.
>
> **Environment Introduction:** Scenario A is a small map (square boundary), scenario B (square boundary) and scenario C (rectangle boundary) are big maps. The observation space of each scenario is the same, but the scene scale, boundary shape, and obstacle placement are different. [See anonymous link for visualization of the tasks](https://anonymous.4open.science/r/ICML_reviewer_3-0622/README.md).
>
> **Action space:** Discretization direction: left, right, forward, back; Continuous action (Speed): the number of squares to walk in the corresponding direction (1 to 4). The continuous action space is [0.5,4.5], and the environment will discretize the continuous actions to ensure that they are in integer units: [0.5,1.5) is one step, [1.5,2.5) is two, and so on.
>
> We evaluate all methods in the above three scenarios according to the experimental settings in this paper. The results in Table 1 show that our method outperforms all baselines in three complex scenarios, which further proves the effectiveness of our method. The poor performance of the current SOTA method, i.e. HyAR-TD3, indicates the inefficiency of its exploration ability.
>
> Table 1. Win rate (Average of 10 runs):
> | Method      | Scenario A (%)| Scenario B (%)| Scenario C (%)|
> | :-: | :-: | :-: | :-: |
> | HyAR-TD3|$40$|$20$|$30$|
> | HPPO |$30$|$0$|$30$|
> | PDQN |$10$ |$0$|$10$|
> | HHQN |$30$ |$10$|$20$|
> | **Ours** |$60$ |$40$|$50$|
>
> **Q1.2: Algorithm 1 is confusing, requiring traversing all sub-regions first, then doing MCTS and eventually, training the transformer. This suggests that a very good initialization of the value and policy networks is needed, with samples covering most of the interesting parts of the state-action space.**
>
> Thanks for pointing out the vague part, we will explain it in detail and add additional experiments in the new version. It is not necessary to traverse all sub-regions before performing MCTS, the reason we do this is that we found in the experiment process that traversing the sub-regions first can improve the learning speed of the policy. The policy without traversing all sub-regions can also achieve a similar level of performance (outperforming the other baselines) with the help of MCTS's powerful ability to balance exploration and exploitation.
>
> We expose the performance of the methods. Method 1: traversing all sub-regions first. Method 2:  without traversing all sub-regions first and directly doing MCTS. The results in Table 2 show that both methods perform at similar levels, with method 1 performing slightly better on Chase. Table 3 shows the convergence speeds of the two methods, "traversing all sub-regions first" indeed helpful to improve the exploration efficiency of the policy and improve the learning speed.
>
> Table 2. Performance (Average of 5 runs):
> | Method      | Chase | Move| Hard Goal|
> | :-: | :-: | :-: | :-: |
> | Method 1|$0.81\pm 0.08$|$0.97\pm 0.04$|$0.58\pm 0.13$|
> | Method 2|$0.77\pm 0.05$|$0.95\pm 0.08$|$0.56\pm 0.17$|
>
> Table 3. Learning speed (Average of 5 runs):
> | Method      | Chase | Move| Hard Goal|
> | :-: | :-: | :-: | :-: |
> | Method 1|$2h12min$|$1h26min$|$8h41min$|
> | Method 2|$2h48min$|$2h05min$|$10h13min$|

---

### Official Review · Reviewer_4D6k · 2024-03-25

**Q2-1 Originality-Novelty:** 3
**Q2-2 Correctness-Technical Quality:** 3
**Q2-5 Clarity Of Writing:** 3

**Q1 Summary And Contributions:**

The paper considers the problem of RL with hybrid action space, where the action consists of a discrete part and a continuous part. The paper proposes to discretize continuous actions into bins, and use MCTS to find the subregion where the action is promising, and search for good continuous actions within the subregion. Additionally, a transformer architecture is proposed to encode the state, discrete action and subregion for the Q function. Experimental results show that the proposed approach can out-perform state-of-the-art algorithms in various simulation experiments.

**Q2-3 Extent To Which Claims Are Supported By Evidence:**

4: Excellent: all claims are supported by very convincing evidence (in the form of comprehensive experimental evaluation, rigorous mathematical proofs, detailed (pseudo-)code, precise references, well-motivated and realistic assumptions) and the authors deliver what they promise.

**Q2-4 Reproducibility:**

3: Good: key resources (e.g. proofs, code, data) are available and key details (e.g. proofs, experimental setup) are sufficiently well-described for competent researchers to confidently reproduce the main results.

**Q3 Main Strengths:**

1. The proposed approach is technically sound and novel to my knowledge.
2. The research problem is well-motivated, with concrete examples.
3. The explanation of the proposed approach is in general clear, with nicely-made figures illustrating algorithms and network architectures.
4. The experimental results are quite thorough. The proposed approach is compared with multiple baselines and is shown to out-perform them in multiple scenarios consistently.
5. The ablation study shows sensitivity of the approach to the choice of number of subregions, and validates the choice of network architecture.

**Q4 Main Weakness:**

1. It would be nice if the authors can spend more space explaining how the Q values are propagated in Section 3.1 (4). How is $V_{new}$ computed? Are Q values also propagated to the previous time step like other MCTS approaches? How are $\mathbb X_{kw}$ and $\mathbb{Q}_{kw}$ used (are they paired or just two sets)? How does the algorithm handle stochastic dynamics?
2. It seems like the discussions in the paper are mostly focused on having only one discrete component and one continuous component in the action space. It would be nice if the authors can discuss how the algorithms can be adapted to handle scenarios with multiple discrete components and multiple continuous components.

**Q5 Detailed Comments To The Authors:**

Apart from the main weaknesses mentioned above, here are some additional comments/questions:
1. What would happen if a subregion contains mostly bad actions but also some good actions (for example, in Figure 1C, the average Q for the third subregion is actually the lowest). Would MCTS consider this subregion as bad subregion (with low $Q_r$)?
2. In Section 3.1, "continuous parameter space $\mathcal X_k$ marked with rounded rectangles in light *origin*" -> "continuous parameter space $\mathcal X_k$ marked with rounded rectangles in light *orange*"?
3. In Equation (6), is $\mathbb Q_{\hat k, \hat w}$ same as $\mathbb Q_{\hat k\hat w}$?
4. Why are the results of other algorithms not presented for Chase and Hard Goal in Figure 5?
5. Why are the curves noisier (with more spiky intervals) in the ablation studies (Figure 7) compared to Figure 5? What is the environment for Figure 7?
6. It may be good to describe the MLP architecture in more detail in Section 5.2 or in the appendices.

**Q9 Complying With Reviewing Instructions:**

Yes

---

> ### Author Rebuttal · Authors · 2024-04-07
>
> ## Part 1
> We are deeply grateful for your recognition of our paper's motivation, performance, and potential academic impact. Your positive feedback is highly encouraging. Such feedback motivates us to further our research in this promising field. Thank you for your thorough and constructive review.
> ### Weakness
> **W1: It would be nice if the authors can spend more space explaining how the Q values are propagated in Section 3.1 (4). How is $V_{new}$ computed? Are Q values also propagated to the previous time step like other MCTS approaches? How are $`\mathcal{X}_{kw}`$ and $`\mathcal{Q}_{kw}`$ used (are they paired or just two sets)? How does the algorithm handle stochastic dynamics?**
>
> Thanks for your suggestion, we will cover this part in more detail in the new version.
>
> *How is $V_{new}$ computed?*
>
> Following the Q-value calculation process in the mainstream DRL algorithm, we take the current state, the selected discrete action, and the selected continuous subregion as the input of the Q-value network and output the state-action Q-value as $V_{new}$. The reason why we choose Q value as $V_{new}$ is that the numerical value of Q value directly represents the reward expectation of the selected hybrid action at the current state, which can be used as a measure of the nodes in MCTS. With the increased training steps, the Q network will be continuously optimized to achieve a stable state-action evaluation, which helps MCTS eliminate the suboptimal policy caused by the inaccurate evaluation of node value in stochastic dynamic scenarios.
>
> *Are Q values also propagated to the previous time step like other MCTS approaches?*
>
> Different from other works that directly use MCTS for trajectory path planning, we only use MCTS to make decisions in the hierarchical action space according to the current state and do not need to consider the historical state information, so we do not need to propagate to the previous time step.
>
> *How are $\mathbb{X}_{kw}$ and $`\mathbb{Q}_{kw}`$ used:*
>
> These two sets are paired, and each subregion maintains two such sets. After a complete sampling, the selected continuous action will be added to the set of candidate actions $\mathbb{X}_{kw}$, at the same time, the Q value corresponding to the sampled action is added to $`\mathbb{Q}_{kw}`$.
>
> After sampling the discrete action $\hat{k}$ and subregion$\hat{w}$ via MCTS, we select the parameter action with the highest Q-value previously sampled in subregion $\hat{w}$ under discrete action $\hat{k}$ as the final parameter action.
>
> *How does the algorithm handle stochastic dynamics?*
>
>  Your question is very in-depth.
> - We use the Q-value as the node value in MCTS. Compared with other traditional node value optimization methods, using Q values has an advantage: Q values can be continuously optimized. As the number of training steps and sampling increases, the critic is optimized by TD-learning according to mainstream DRL theory. This ensures that the nodes of MCTS are still effectively evaluated in stochastic dynamic scenarios.
> - Change the exploration mode of MCTS. We used the traditional discounted UCB in the main experiment. When faced with random dynamic scenarios, we can replace traditional UCB with UCB-V [1], which is more effective in stochastic dynamics scenarios. The main idea of UCB-V is to add a variability estimation term based on UCB to improve the modeling ability of uncertain environments.
>
> We use the most unstable scenario in our paper, i.e. Chase, to verify the boost brought by UCB-V to our method. In Chase, the agent needs to capture the enemy, which is controlled by a simple algorithm and the enemy's motion is difficult to predict.
>
> Table 1. Performance (Average of 3 runs):
> | Method      | Chase |
> | :-----------: | :-----------: |
> | Hybrid CtrlFormer with disconted-UCB|$0.74\pm 0.14$|
> | Hybrid CtrlFormer with UCB-V  |$0.88\pm 0.06$|
>
> [1]: Mukherjee, Subhojyoti, et al. "Efficient-ucbv: An almost optimal algorithm using variance estimates." Proceedings of the AAAI Conference on Artificial Intelligence. Vol. 32. No. 1. 2018.

---

### Official Review · Reviewer_ZDMu · 2024-03-25

**Q2-1 Originality-Novelty:** 3
**Q2-2 Correctness-Technical Quality:** 3
**Q2-5 Clarity Of Writing:** 3

**Q1 Summary And Contributions:**

This paper studies the problem of hybrid action control, where action spaces are formed in hierarchy with high level discrete actions and low level regions of continuous actions. The authors propose Hybrid Control Transformer (Hybrid CtrlFormer) as shown in Algorithm 1. The key idea is to use MCTS to sample discrete actions and subregions of continuous actions at high levels, and then learn continuous policy networks at low levels. Causal Transformers are used to learn policies and Q values of for hybrid actions. Experiments conducted on several tasks (Hard Goal, Catch Point, Move, Platform, Chase) show that the proposed Hybrid CtrlFormer outperforms 5 baselines (PADDPG, PDQN, HPPO, HHQN, HyAR). Ablation studies also verify the choices of segmenting subregions of continuous actions, the use of Transformers and MCTS.

**Q2-3 Extent To Which Claims Are Supported By Evidence:**

3: Good: the main claims are supported by convincing evidence (in the form of adequate experimental evaluation, proofs, (pseudo-)code, references, assumptions).

**Q2-4 Reproducibility:**

3: Good: key resources (e.g. proofs, code, data) are available and key details (e.g. proofs, experimental setup) are sufficiently well-described for competent researchers to confidently reproduce the main results.

**Q3 Main Strengths:**

1. Hybrid action control is an important problem. The study is well motivated.
2. The proposed solutions are reasonable and novel to my knowledge.
3. Experimental results and ablation studies support the designs.

**Q4 Main Weakness:**

1. The major weakness is perhaps the tasks used to evaluate the effectiveness of the proposed methods are not challenging. Checking how the proposed method behave on a more complicated environment would make the proposal much more convincing.

**Q5 Detailed Comments To The Authors:**

1. There is an ablation experiment on the effect of segmenting subregions of continuous actions. However, it does not tell how the hierarchy of discrete-subregion-continuous is determined. I think choosing the hierarchy does matter for the performance of the method. Do you have to manually figure out the hierarchy case by case, or how do you determine the hierarchy if it is not naturally given in the problem?

**Q9 Complying With Reviewing Instructions:**

Yes

---

> ### Author Rebuttal · Authors · 2024-04-07
>
> We are immensely grateful to you for recognizing the importance and novelty of our approach. Your valuable suggestions inspire us to improve our work further. If you think the following response addresses your concerns, we would appreciate it if you could kindly consider raising the score.
>
> ### Weakness
> **W1: The major weakness is perhaps the tasks used to evaluate the effectiveness of the proposed methods are not challenging. Checking how the proposed method behave on a more complicated environment would make the proposal much more convincing.**
>
> Thanks for your valuable questions! We will add validation on complex scenarios in the new version. We use minigrid to construct three  difficult scenarios that share the same task logic: The agent must take the key to open the door and go to the goal point. The reason why these task is difficult is that even in the setting of a single discrete space, the mainstream DRL algorithms such as TD3 and PPO. cannot completely solve these tasks due to the difficulty of exploration. These three scenes have the same targets, but different map sizes and boundary ranges.
>
> **Environment Introduction:** Scenario A is a small map (square boundary), scenario B (square boundary) and scenario C (rectangle boundary) are big maps. The observation space of each scenario is the same, but the scene scale, boundary shape, and obstacle placement are different. [See anonymous link for visualization of the tasks](https://anonymous.4open.science/r/ICML_reviewer_3-0622/README.md).
>
> **Action space:** Discretization direction: left, right, forward, back; Continuous action: Speed, the number of squares to walk in the corresponding direction (1 to 4). The continuous action space is [0.5,4.5], and the environment will discretize the continuous actions to ensure that they are in integer units: [0.5,1.5) is one step, [1.5,2.5) is two, and so on.
>
> We evaluate all methods in the above three scenarios according to the experimental settings in this paper. The results in Table 1 show that our method outperforms all baselines in three complex scenarios, which further proves the effectiveness of our method. The poor performance of the current SOTA method, i.e. HyAR-TD3, indicates the inefficiency of its exploration ability.
>
> Table 1. Win rate (Average of 10 runs):
> | Method      | Scenario A (%)| Scenario B (%)| Scenario C (%)|
> | :-----------: | :-----------: | :------------: | :-----------: |
> | HyAR-TD3|$40$|$20$|$30$|
> | HPPO |$30$|$0$|$30$|
> | PDQN |$10$ |$0$|$10$|
> | HHQN |$30$ |$10$|$20$|
> | **Ours** |$60$ |$40$|$50$|
>
> ### Detailed Comments
> **C1: There is an ablation experiment on the effect of segmenting subregions of continuous actions. However, it does not tell how the hierarchy of discrete-subregion-continuous is determined. I think choosing the hierarchy does matter for the performance of the method. Do you have to manually figure out the hierarchy case by case, or how do you determine the hierarchy if it is not naturally given in the problem?**
>
> Thanks for your insightful questions, we will improve the presentation of this part in the new version. Our hierarchical structure is task-agnostic. We only need to specify the number of subregions in the continuous action space. According to each discrete action and its corresponding number of continuous action dimensions, the hierarchy is automatically generated.
>
> For any task, the hierarchical action architecture can be designed as follows: the first level is all discrete actions, the second level is continuous action subregions corresponding to each discrete action and the third level is continuous action in subregion. Notably, to achieve task-agnostic, we use sequential splitting to partition the continuous action space, that is, the user only needs to set the desired number of subspaces, and each continuous action space is uniformly split into c subregions.

---

### Meta-Review · Area_Chair_R9rG · 2024-04-16

The paper addresses the problem of hybrid decision making where the set of actions include both discrete and continuous variables. The key novelty is the introduction of a Transformer-based MCTS to split the search space more efficiently and facilitate sampling.

The reviewers agree that the paper makes an interesting and valuable contribution. They also requested a few clarifications such as details on the computation of V_new and Q, performance under stochastic dynamics, scalability, etc. After reading the reviews and the rebuttal by the authors, I believe the paper presents a useful method with interesting applications. I would encourage the authors to include more challenging tasks in the experiments to better demonstrate the benefits of the approach.